# Performance of Risk Scores in Predicting Infective Endocarditis in Patients with *Staphylococcus aureus* Bacteraemia in a Prospective Asian Cohort

**DOI:** 10.3390/jcm13102947

**Published:** 2024-05-16

**Authors:** Jinghao Nicholas Ngiam, Matthew Chung Yi Koh, Sophia Archuleta, Dale Fisher, Louis Yi-Ann Chai, Ching-Hui Sia, William K. F. Kong, Paul Anantharajah Tambyah

**Affiliations:** 1Division of Infectious Diseases, National University Hospital, National University Health System, 1E Kent Ridge Rd, NUHS Tower Block, Level 10, Singapore 119228, Singapore; matthewkoh91@gmail.com (M.C.Y.K.);; 2Department of Medicine, Yong Loo Lin School of Medicine, National University of Singapore, Singapore 119077, Singapore; 3Department of Cardiology, National University Heart Centre Singapore, National University Health System, 1E Kent Ridge Rd, NUHS Tower Block, Level 9, Singapore 119228, Singapore; 4Infectious Diseases Translational Research Programme, Department of Medicine, Yong Loo Lin School of Medicine, National University of Singapore, Singapore 119077, Singapore

**Keywords:** *Staphylococcus aureus* bacteraemia, infective endocarditis, risk score, mortality

## Abstract

**Background**: Several risk scores have been derived to predict the risk of infective endocarditis (IE) amongst patients with *Staphylococcus aureus* bacteraemia (SAB), which helps to guide clinical management. **Methods**: We prospectively studied 634 patients admitted with SAB. The cohort was stratified into those with or without IE, and the PREDICT Day 1, Day 5 and VIRSTA scores were tabulated. Area under the receiver operating characteristic (AUC) curves were constructed to compare the performance of each score. **Results**: Of the 634 patients examined, 36 (5.7%) had IE. These patients were younger (51.6 ± 20.1 vs. 59.2 ± 18.0 years, *p* = 0.015), tended to have community acquisition of bacteraemia (41.7% vs. 17.9%, *p* < 0.001), and had persistent bacteraemia beyond 72 h (19.4% vs. 6.0%, *p* = 0.002). The VIRSTA score had the best performance in predicting IE (AUC 0.76, 95%CI 0.66–0.86) compared with PREDICT Day 1 and Day 5. A VIRSTA score of <3 had the best negative predictive value (97.5%), compared with PREDICT Day 1 (<4) and Day 5 (<2) (94.3% and 96.6%, respectively). **Conclusions**: Overall, the risk scores performed well in our Asian cohort. If applied, 23.5% of the cohort with a VIRSTA ≥ 3 would require TEE, and a score of <3 had an excellent negative predictive value.

## 1. Introduction

With the increasing prevalence of *Staphylococcus aureus* and antibiotic resistance, *Staphylococcus aureus* bacteraemia (SAB) is a growing threat in Asia with high mortality and morbidity [1]. Furthermore, specific to Singapore, a rising incidence of SAB and infective endocarditis has been observed in intravenous drug abusers, with the use of buprenorphine [2]. The bacterium itself has significant virulence, being able to seed in multiple foci and can cause infective endocarditis (IE) in patients with structurally normal cardiac valves or without any predisposing risk factors. It also has the propensity to relapse and recur at metastatic sites if inadequately treated [3]. In recent years, *Staphylococcus aureus* has been demonstrated to be the commonest cause of IE [4]. In many ways, the increased incidence of SAB, and in particular, healthcare-associated acquisition of the bacteraemia, appeared to be related to the higher usage of central lines and intracardiac devices [5]. The lines and devices act as foreign material that allow for bacterial seeding. Furthermore, the breach of the skin barrier also further poses a significant risk factor for the development of SAB [6].

A significant proportion of patients with SAB may be complicated by IE, as defined the modified Duke criteria [7,8]. Several modalities had been identified to evaluate the possibility of IE, including echocardiography (transthoracic, transoesophageal), cardiac computed tomography (CT) and positron emission tomography (PET) [9]. Cardiac CT is less accurate than ultrasonography in identifying valvular vegetations and perforations. While transoesophageal echocardiography (TEE) may be more sensitive for IE than transthoracic echocardiography (TTE), it is more costly, more resource demanding and semi-invasive [10,11].

As such, the resources required to evaluate IE in patients with SAB may be significant. Some modalities are not readily accessible in resource-limited settings as well. In order to circumvent these limitations, several strategies have since been employed to risk-stratify patients who would benefit from further detailed evaluation (e.g., TEE). This would allow for the best utilisation of resources in the evaluation of patients with SAB. Essentially, risk scores aimed to identify patients at low risk who may be treated with a shorter duration of antibiotics, while the remaining moderate-to-high-risk patients would benefit from further invasive or more detailed evaluation to exclude IE. This important distinction exists as patients with IE require longer antibiotic treatment durations, and in certain cases, may also benefit from surgical management.

Some of these previously described tools are risk scores, such as “Predicting Risk of Endocarditis using a Clinical Tool (PREDICT)” and “VIRSTA” [12,13]. They have been successfully employed in various clinical settings as an algorithm to identify patients with low pre-test probability of IE. The PREDICT Day 1, Day 5 and VIRSTA clinical prediction tools to evaluate the risk of IE in patients with SAB are summarised in Appendix A. This would therefore reduce the need for TEE overall and prioritise the patients who are at the highest risk for IE. While these scores have been evaluated and validated in Western populations, our study aimed to assess the real-world feasibility of applying these prediction tools in a clinical setting of patients admitted with SAB, and to subsequently evaluate and compare the performance of these risk scores in predicting IE in our Asian population with SAB.

## 2. Materials and Methods

### 2.1. Study Population

In order to do so, we prospectively studied 634 consecutive hospitalised patients with SAB in a tertiary academic centre in Singapore. This study was conducted over a period of 7 years, from 2008 to 2014. All of the cases had the diagnosis of SAB confirmed with positive blood cultures. They were all recruited prospectively within 24–48 h of the positive blood culture. Written informed consent was obtained from each patient prior to enrolment. In the context of this prospective cohort study, the management of each patient, such as antibiotic choice, duration and need for serial imaging or further evaluation, was left to the discretion of the primary managing physician.

For the purposes of this study, each enrolled patient with SAB had a prospective follow-up. For each subject, data were collected on their baseline demographics and background, laboratory findings and clinical characteristics. Community-acquired bacteraemia was defined as a positive blood culture obtained within 48 h of patient admission. These clinical characteristics included the presence of persistent bacteraemia, metastatic foci of infection and mortality. The primary outcome measure was the development and diagnosis of IE during the index hospital admission. Prospective follow-up for the primary outcome for each patient was carried out until the point of discharge, or in-hospital mortality. In addition, secondary outcome measures, including the recurrence of SAB within 1 year and all-cause hospital readmissions within 1 year, were tabulated.

### 2.2. Statistical Analyses

The study population was subsequently divided into those with or without IE. Infective endocarditis was defined by the modified Duke criteria [14]. A patient was deemed to have IE if the patient fulfilled either two major criteria, one major and three minor criteria, or all five minor criteria for IE. These criteria were classified by the primary managing clinical team for each patient and, where requested, in consultation with an Infectious Diseases specialist team. A multidisciplinary endocarditis team was not always available in our clinical setting, but the clinical, microbiological and imaging criteria were also reviewed by the study team (JNN and MCYK) to adjudicate if each patient fulfilled or did not fulfil the criteria for the diagnosis of IE.

We subsequently calculated the PREDICT Day 1, Day 5 and VIRSTA scores for each patient in our cohort. Clinical, microbiological and laboratory parameters, such as the presence of pre-existing native valvular disease, previous infective endocarditis, the presence of implantable cardiac devices, community or healthcare acquisition of bacteraemia, persistently positive blood cultures and other embolic phenomena, were tabulated in order to calculate the scores for each individual patient. A detailed breakdown of the items of each clinical score and its calculation is shown in Appendix A. A positive PREDICT Day 1 score was defined as 4 points or more, a positive PREDICT Day 5 score was 2 points or more, and a positive VIRSTA score was 3 points or more. Each patient then also had a prospective follow-up during the hospital admission for the primary outcome, which was the presence or development of IE.

To compare differences in characteristics between the two groups, we employed a t-test for continuous parameters. For these continuous parameters, a histogram was first constructed to check that the variable was normally distributed. For categorical parameters, chi-squared tests were performed. The continuous parameters were presented as mean (±one standard deviation). The categorical parameters were presented as frequencies (and percentages). To evaluate the performance of each risk score, we calculated the area under the receiver operating characteristic (AUC) curves.

All data analyses were performed using SPSS version 20.0 (SPSS, Inc., Chicago, IL, USA). For the analyses, a *p*-value less than 0.05 was deemed significant. This study had been approved by the National Healthcare Group Domain Specific Review Board (DSRB, 2006/00274) prior to its conduct. This study was conducted in accordance with the principles laid out by the Declaration of Helsinki. We ensured that all collected data were anonymised.

## 3. Results

Amongst the consecutive 634 patients with SAB, 36 (5.7%) had IE while the remaining 598 (94.3%) did not. The follow-up period was for the duration of the hospital stay, for an average of 23 ± 21 days. The patients with IE were younger with a mean age of 51.6 ± 20.1 years old compared with 59.2 ± 18.0 years old in those without IE (*p* = 0.015). Furthermore, those with IE were more likely to have community-acquired SAB (41.7% vs. 17.9%, *p* < 0.001), as well as having risk factors such as intravenous drug usage (30.6% vs. 2.7%, *p* < 0.001) and underlying structural heart disease (11.1% vs. 2.7%, *p* = 0.005) compared to those without IE (Table 1).

IE was associated with persistent fever beyond 72 h (19.4% vs. 5.5%, *p* = 0.001) and persistent bacteraemia beyond 72 h (19.4% vs. 6.0%, *p* = 0.002). Vascular/embolic phenomena were also observed more commonly in the group with IE (38.9% vs. 3.2%, *p* < 0.001). There was greater evidence of inflammation in the IE group, with a higher C-reactive protein (CRP) of 178.6 ± 73.1 mg/L compared with 141.4 ± 108.3 mg/L in the group without (*p* = 0.143). Similarly, the group with IE had a higher total white cell count of 17.0 ± 7.7 × 109/L compared with 14.1 (±7.3) × 109/L in the group without (*p* = 0.04). We did not demonstrate a significant difference in in-hospital mortality between the IE and non-IE groups (8.3% vs. 5.7%, *p* = 0.459).

In our local context, the VIRSTA score of ≥3 performed the best in identifying patients with IE, with an area under the receiver operating characteristic curve of 0.76 (95%CI 0.66–0.86), compared with the PREDICT Day 1 score of 0.61 (95%CI 0.51–0.71) and a PREDICT Day 5 score of 0.65 (95%CI 0.58–0.78) (Figure 1). At a cut-off of ≥3, the VIRSTA score had 66.7% sensitivity and 79.1% specificity, while a score of 2 or less was associated with a 97.5% negative predictive value. Comparatively, the PREDICT Day 5 score ≥ 2 had 58.3% sensitivity and 71.4% specificity, and a score of 1 or less had a 96.6% negative predictive value (Table 2).

## 4. Discussion

The utilisation of risk scores to predict IE in patients with SAB is of paramount importance. It helps to guide further investigations, identifying higher-risk patients who would benefit from TEE evaluation, and low-risk patients who may be managed with a shorter duration of antibiotics. These considerations would be critical in resource-limited settings where advanced imaging modalities may not be easily available, or where there may be limited access to TEE. In addition, in many clinical settings, patients may also be apprehensive about proceeding with invasive testing. In these clinical scenarios, the use of such risk scores would be especially helpful in guiding clinicians on the management of patients with SAB, by identifying those at the highest risk of IE who would benefit the most from invasive testing [15].

Our study demonstrated that the risk scores performed reasonably well in predicting patients who had IE amongst those with SAB. In particular, the VIRSTA score did better than the other commonly used PREDICT Day 1 and Day 5 scores. Most strikingly, the VIRSTA score appeared to be most useful when the scores were low. At scores of 2 or less, it had an excellent negative predictive value of 97.5%. This would have the important implication that these patients could be managed with antibiotics alone, without the need to pursue more invasive investigations, such as TEE. In fact, if the VIRSTA score had been applied uniformly to our study population, approximately 149 (23.5%) patients would have been recommended for TEE, while the remaining 76.5% could have been managed expectantly without further invasive testing. These would be particularly important in appropriately stratifying resources to allocate TEE examinations to patients at the highest risk for developing IE.

Although TEE is deemed an invasive procedure, it is important to note that in our local context in Singapore, TEE is considered a highly safe procedure. In a retrospective study, no mortality was noted in 901 patients, with major complications seen in only 5 (0.6%) patients within the study population [16]. Nevertheless, it is still resource intensive, and the existing risk scores would be helpful in identifying patients who would benefit most from this procedure.

The findings support and validate the use of widely available risk scores for the local and Asian context of patients with SAB. However, it was important to note that, in our Singapore cohort, the proportion of patients with IE was low, with only 5.7% of the patients with SAB having IE. In a Danish study that prospectively recruited patients with SAB, 22% had IE, and this was elevated to 38% in those with prosthetic cardiac valves or devices [17]. This is of particular concern, since the presence of IE has been associated with 2.8 times increased odds of mortality in patients with SAB [18]. Several factors may contribute to the lower observed incidence of IE in our cohort. For example, a significant proportion of our patients with SAB were hospital acquired (~80%). This may be associated with a relatively shorter duration of bacteraemia and, consequently, a lower risk of developing IE [19]. When bacteraemia is acquired in the hospital, the identification of the causative organism and the initiation of appropriate antibiotic therapy is often rapid. Conversely, patients who acquire the infection from the community may have a longer duration of fever and bacteraemia, with the initiation of effective antibiotics later in their disease course, and may thus consequently be at a higher risk of developing IE [20].

In addition, a significant proportion of patients in our local context had acquired their bacteraemia as a consequence of central lines. These included patients with end-stage kidney disease who required catheter insertion for vascular access for haemodialysis. In fact, approximately 20% of our cohort had been on haemodialysis. In such patients, the offending catheter was often quickly removed, which led to the rapid resolution of the bacteraemia [21]. The shorter duration of bacteraemia in these patients, with effective and timely control of the infective source would thus in turn lead to a lower likelihood of the development of IE.

Another finding to note is that the in-hospital mortality in our cohort of patients with SAB was also very low. The overall in-hospital mortality was 5.8% and did not differ significantly between the group with IE and those without. The low mortality may in part be associated with a lower proportion of patients with IE, as discussed above, as IE, along with other metastatic foci of infection, has been consistently associated with a higher risk of mortality in the context of patients with SAB [22]. Of note, we did not control or adjust for the length of antibiotic therapy and need for surgery in our cohort of patients. When IE was appropriately identified and treated, the resultant mortality may be consequently lower and comparable to those without IE. This further highlights the importance of identifying IE and metastatic sites of infection in such patients, to obtain adequate source control and improve clinical outcomes in patients with IE.

Additionally, there are also other factors that may have led to a lack of observable difference in mortality between the IE and non-IE groups. In fact, another factor that may contribute to overall lower mortality in patients with SAB IE may be unique to the Singapore context. Of note, in the years in which this study was conducted, intravenous drug abuse with buprenorphine was prevalent in the local community. This led to several cases of community-onset SAB [23]. In our study cohort, IVDU represented a small but significant proportion (~4%) of the study population, with a large proportion of them developing IE (up to 30%). Although this group of patients were more likely to develop IE, they had also been young and relatively fit, with few medical co-morbidities. As such, with adequate and appropriate duration of therapy, they had been less likely to experience mortality from SAB. Importantly, despite the lower incidence of IE and mortality from SAB in our local setting, the risk scores examined still performed reasonably well.

Despite a relatively low proportion of patients with IE in our cohort, our findings demonstrate concordance with the existing literature on risk factors for the development of IE in patients with SAB [24,25]. The important risk factors for the development of IE remain the presence of persistent fevers while on therapy, prolonged duration of bacteraemia and an existing cardiac implantable device or prosthesis [24,25]. These are also important components of the VIRSTA and PREDICT scores. The presence of persistent fevers and bacteraemia suggests an endovascular source of infection that results in high-grade bacteraemia with the organism being consistently isolated on repeated cultures. The presence of an in-dwelling cardiac device forms a nidus for which the bacteria can seed and lead to the development of IE. As such, when these factors are present, the risk of IE is significantly higher. These are the patients who would benefit most from invasive testing with TEE to evaluate for IE in the context of SAB.

## 5. Limitations

This was a moderately sized but single-centre study that examined the risk of IE in patients with SAB. We did not perform sample size estimation for this prospective cohort as we had not aimed to demonstrate a difference in clinical outcomes between two interventions. Instead, we recruited patients admitted with SAB within the study period and followed up with them prospectively for the presence of IE and clinical outcomes. Some characteristics of the cohort, like a higher proportion of hospital-acquired bacteraemia and the lower mortality, may be unique to the local context and limit the generalisability of the findings. Furthermore, we also had a comparatively low proportion of patients with prosthetic cardiac devices, implants or valves (6.8%), of which only three patients had prosthetic cardiac valves. Nevertheless, we still prospectively followed up with each patient to characterise the clinical outcomes, and there were no patients that had been lost to follow-up. Very few patients had TEE (3.3%), and most of the vegetations from IE had been detected on TTE. In the local context, several patients declined invasive investigations such as TEE. If more patients had TEE, it is possible that more cases of IE would be picked up and that we underestimated the incidence of IE in our study cohort. However, the mortality rates were quite low in our population, with a relatively short duration of fever and bacteraemia, which were both consistent with a lower rate of IE. Another risk score, the POSITIVE score, has also been used to determine risk for IE in patients with SAB [26]. This score heavily relied on the time-to-positivity of blood cultures. We did not collect these data and thus could not validate if this score was useful in our local context.

Another important consideration in the diagnosis of IE is patients who had received antibiotics prior to clinical presentation. Pre-treatment with antibiotics may result in negative cultures that may ‘mask’ the diagnosis of IE. However, all of our patients had positive cultures for *S. aureus* in this cohort. Nevertheless, the initial cardiac imaging with echocardiography may have been equivocal, necessitating serial TTE or TEE for the diagnosis of IE. In the context of this prospective cohort study, the decision pertaining to the need for repeat imaging in this setting was left to the discretion of the managing physician. We had not been able to quantify the proportion of patients who received antibiotics prior to admission and also had not been able to quantify the number of patients who had initially equivocal imaging results that necessitated repeat imaging. This would be an important limitation to consider, as it may have resulted in a more delayed diagnosis of IE.

Importantly, this study had focused on evaluating the risk factors for the development of IE in patients with SAB. However, we only considered the development of IE during the index admission for SAB and did not follow each patient longitudinally, as IE may have developed on subsequent hospital admissions. The average follow-up period was the length of hospitalisation (23 ± 31 days). However, we did examine data on the recurrence of bacteraemia and all-cause readmissions within a year from the episode of SAB, and these did not differ significantly between the groups.

Furthermore, we also did not examine risk factors for mortality in our cohort. The patients with IE were not always managed by a multidisciplinary endocarditis team in our clinical setting. We also did not systemically record if the patients had undergone valve replacement surgery as part of IE management, or if they had indwelling catheters or cardiac devices extracted as part of the management of SAB. Treatment decisions were also left to the discretion of the managing physicians. Nevertheless, mortality was relatively low at 5.8%, and we did not demonstrate a difference in mortality between the IE and non-IE cohort. In addition, we did not control or adjust for the duration of antibiotic therapy as well as the need for surgery for IE. It was possible that with appropriate identification and management of IE in patients with SAB, the mortality may be similar to those who did not have IE. The low mortality also limited the ability to perform comprehensive analyses on this specific outcome as well. Evaluating the presence of a multidisciplinary endocarditis team, the frequency of valve replacement, device extraction and catheter removal would be critical when evaluating the risk of mortality in patients with SAB and may be an important subject for future studies. Although we could not demonstrate the impact of the risk scores on in-hospital mortality, we believe that our findings provide valuable insights into the role of existing risk scores in predicting IE in a relatively ‘low-risk’ population of patients with SAB.

## 6. Conclusions

Overall, the risk scores performed well in our Asian cohort, despite a lower incidence of IE and in-hospital mortality. The VIRSTA score performed better than the PREDICT score. If the VIRSTA score was applied, 23.5% of SAB patients would require TEE, while the remaining cohort with a score of <3 could be managed expectantly, with a negative predictive value of 97.5%.

## Figures and Tables

**Figure 1 jcm-13-02947-f001:**
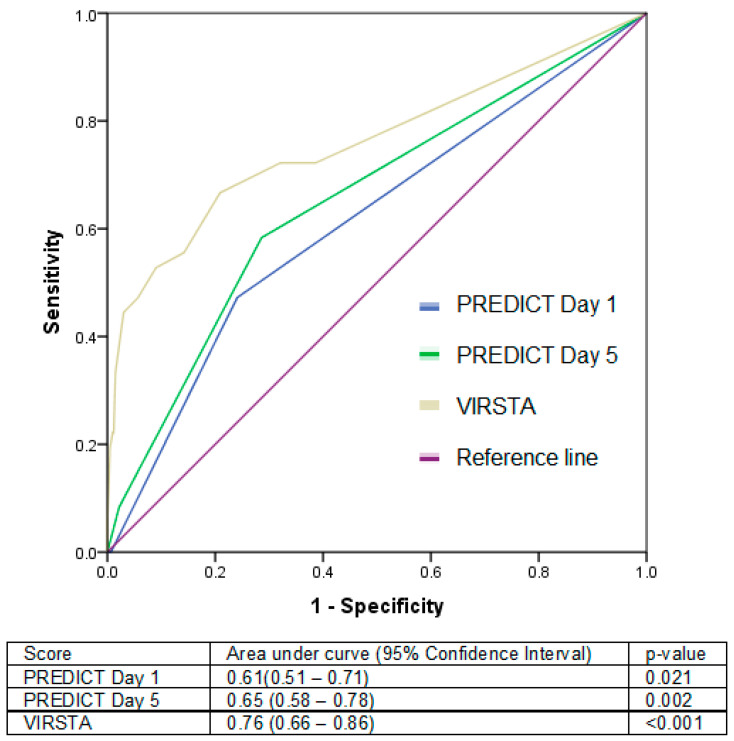
Receiver operating curves comparing performance of each risk score in predicting infective endocarditis in patients with *Staphylococcus aureus* bacteraemia.

**Table 1 jcm-13-02947-t001:** Characteristics of patients with *Staphylococcus aureus* bacteraemia, stratified by patients with or without infective endocarditis.

Parameter	Overall Cohort (*n* = 634)	Infective Endocarditis (*n* = 36)	No Infective Endocarditis (*n* = 598)	*p*-Value
Age (years)	58.7 (±18.2)	51.6 (±20.1)	59.2 (±18.0)	0.015
Sex (male)	403 (64.1%)	26 (72.2%)	377 (63.6%)	0.294
Community-acquired bacteraemia (within 48 h of admission)	122 (19.2%)	15 (41.7%)	107 (17.9%)	<0.001
Methicillin-resistant *Staphylococcus aureus*	165 (26.0%)	9 (25.0%)	156 (26.1%)	0.885
Intravenous drug use	27 (4.3%)	11 (30.6%)	16 (2.7%)	<0.001
Structural heart disease	20 (3.2%)	4 (11.1%)	16 (2.7%)	0.005
Prosthetic cardiac device/implant or valves	43 (6.8%)	2 (5.6%)	41 (6.9%)	0.763
Presence of indwelling vascular catheter	118 (18.6%)	3 (8.3%)	115 (19.2%)	0.103
Recent surgery	138 (21.8%)	5 (13.9%)	133 (22.2%)	0.238
Human immunodeficiency virus infection	5 (0.8%)	0 (0.0%)	5 (0.8%)	0.999
Diabetes mellitus	255 (40.2%)	9 (25.0%)	246 (41.1%)	0.055
End-stage kidney disease	145 (22.9%)	2 (5.6%)	143 (23.9%)	0.011
Persistent fever beyond 72 h	40 (6.3%)	7 (19.4%)	33 (5.5%)	0.001
Persistent bacteraemia beyond 72 h	43 (6.8%)	7 (19.4%)	36 (6.0%)	0.002
Total white cell count ×10^9^/L	14.3 (±7.4)	17.0 (±7.7)	14.1 (±7.3)	0.046
C-reactive protein (mg/L)	144.3 (±106.4)	178.6 (±73.1)	141.4 (±108.3)	0.143
Serum albumin (g/dL)	32.3 (±7.6)	29.8 (±7.0)	32.6 (±7.6)	0.161
Serum creatinine (µmol/L)	268.1 (±30.9.5)	143.9 (±253.9)	277.6 (±311.8)	0.086
Vascular or embolic phenomena	33 (5.2%)	14 (38.9%)	19 (3.2%)	<0.001
Bone infection	19 (3.0%)	2 (5.6%)	17 (2.8%)	0.354
Transoesophageal echocardiography	21 (3.3%)	6 (16.7%)	15 (2.5%)	0.001
In-hospital mortality	37 (5.8%)	3 (8.3%)	34 (5.7%)	0.459
Recurrence of SAB within 1 year	3 (0.5%)	0 (0.0%)	3 (0.5%)	0.999
All-cause readmission within 1 year	9 (1.4%)	1 (2.8%)	8 (1.3%)	0.411
Length of stay (days)	23 (±31)	32 (±23)	22 (±32)	0.166
VIRSTA Score	1.5 (±2.5)	5.4 (±4.7)	1.3 (±2.1)	<0.001
VIRSTA Score ≥ 3	149 (23.5%)	24 (66.7%)	125 (20.9%)	<0.001
PREDICT Day 1 Score	0.5 (±0.9)	0.9 (±1.0)	0.5 (±0.9)	0.004
PREDICT Day 1 Score ≥ 4	4 (0.6%)	0 (0.0%)	4 (0.7%)	0.623
PREDICT Day 5 Score	0.7 (±1.0)	1.3 (±1.3)	0.6 (±1.0)	<0.001
PREDICT Day 5 Score ≥ 2	192 (30.3%)	21 (58.3%)	171 (28.6%)	<0.001

**Table 2 jcm-13-02947-t002:** Comparing performance of each risk score at specified cut-offs in predicting infective endocarditis in patients with *Staphylococcus aureus* bacteraemia.

Score	Proportion of Cohort	Sensitivity (%)	Specificity (%)	Positive Predictive Value (%)	Negative Predictive Value (%)
PREDICT Day 1 ≥ 4	4/634 (0.6%)	0.0	99.3	0.0	94.3
PREDICT Day 5 ≥ 2	192/634 (30.2%)	58.3	71.4	10.9	96.6
VIRSTA ≥ 3	149/634 (23.5%)	66.7	79.1	16.1	97.5

## Data Availability

Data may be made available upon reasonable request from the corresponding author.

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
