# Peer review of "Performance of Risk Scores in Predicting Infective Endocarditis in Patients with Staphylococcus aureus Bacteraemia in a Prospective Asian Cohort"

_jcm, 2024, doi:10.3390/jcm13102947_

Round 1

Reviewer 1 Report

Comments and Suggestions for Authors

I am grateful to the editor for the opportunity to review the manuscript by Ngiam et al. “Performance of risk scores in predicting infective endocarditis in patients with Staphylococcus aureus bacteraemia in a prospective Asian cohort.” In this study, the authors examined the capabilities of two risk assessment scales for infective endocarditis in patients with Staphylococcus aureus bacteraemia. The study was conducted on a fairly representative cohort of patients (634 patients) in one of the clinics in Singapore. The authors were able to show that the VIRSTA scale is more suitable for assessing the risk of infective endocarditis in this category of patients, allowing to identify patients who are indicated for semi-invasive TEE (in 23.5% of cases), as well as patients who are not indicated for further examination (with VIRSTA less than 3 points).

Comments when reviewing the manuscript:

1. 1. There is no information in section 2. Materials and Methods about checking quantitative data for normality of distribution. If the distribution is different from normal, it is incorrect to present the results as mean (±1 standard deviation) and compare them using a t-test.

2. In section 2. Materials and Methods, it is advisable to highlight subheadings.

2. The phrase “A p-value of less than 0.05 was considered statistically significant in this study” (lines 100-101) is then repeated again (lines 103-104).

3. There is no study limitation section.

4. Stylistic editing is necessary in the discussion - educational section The discussion begins with the main result obtained by the authors.

5. In the discussion, it is advisable to consider other publications that consider the factors for identifying infective endocarditis with Staphylococcus aureus bacteraemia (for example, ref. 1-2, see below)

6. The study noted no differences in mortality in groups with and without infective endocarditis. This is an interesting observation, but it has not been sufficiently studied by the authors. It is advisable to evaluate factors associated with mortality in patients with Staphylococcus aureus bacteraemia using univariate logistic regression.

References:

1. Heriot GS, Cronin K, Tong SYC, Cheng AC, Liew D. Criteria for Identifying Patients With Staphylococcus aureus Bacteremia Who Are at Low Risk of Endocarditis: A Systematic Review. Open Forum Infect Dis. 2017 Nov 24;4(4):ofx261. doi: 10.1093/ofid/ofx261.

2.Khan UA, Zaidi SH, Majeed H, Lopez E, Tofighi D, Andre P, Schevchuck A, Garcia ME, Sheikh AB, Raizada V, Sheikhar R, Sagheer S. Clinical Risk Factors for Infectious Endocarditis Patients With Staphylococcus Aureus Bacteremia and the Diagnostic Utility of Transesophageal Echocardiogram. Curr Probl Cardiol. 2022 Nov;47(11):101331. doi: 10.1016/j.cpcardiol.2022.101331.

Comments on the Quality of English Language

No comments

Author Response

Ref.:  Ms. No. jcm-2920608

Title: Performance of risk scores in predicting infective endocarditis in patients with Staphylococcus aureus bacteraemia in a prospective Asian cohort

Journal of Clinical Medicine

We thank the Editor for allowing us the opportunity to revise our manuscript and the Reviewers for the important and constructive comments.  We have amended our paper in order to address the points raised by the Reviewer.

In the sections below, each of the points raised is identified and addressed with changes in the revised manuscript.

REVIEWER 1:

I am grateful to the editor for the opportunity to review the manuscript by Ngiam et al. “Performance of risk scores in predicting infective endocarditis in patients with Staphylococcus aureus bacteraemia in a prospective Asian cohort.” In this study, the authors examined the capabilities of two risk assessment scales for infective endocarditis in patients with Staphylococcus aureus bacteraemia. The study was conducted on a fairly representative cohort of patients (634 patients) in one of the clinics in Singapore. The authors were able to show that the VIRSTA scale is more suitable for assessing the risk of infective endocarditis in this category of patients, allowing to identify patients who are indicated for semi-invasive TEE (in 23.5% of cases), as well as patients who are not indicated for further examination (with VIRSTA less than 3 points).

Comments when reviewing the manuscript:

  1. There is no information in section 2. Materials and Methods about checking quantitative data for normality of distribution. If the distribution is different from normal, it is incorrect to present the results as mean (±1 standard deviation) and compare them using a t-test.

We thank the reviewer for this comment. We have included that we checked for normal distribution for continuous variables before performing a t-test (Page 3).

  1. In section 2. Materials and Methods, it is advisable to highlight subheadings.

We have included subheadings for materials and methods (Page 3)

  1. The phrase “A p-value of less than 0.05 was considered statistically significant in this study” (lines 100-101) is then repeated again (lines 103-104).

Thank you for pointing this out. We have removed the repeated line (Page 3).

  1. There is no study limitation section.

We have expanded upon this and included a section on the study limitations (Page 9).

  1. Stylistic editing is necessary in the discussion - educational section The discussion begins with the main result obtained by the authors.

We have rephrased the discussion for greater clarity and focus (Page 7).

  1. In the discussion, it is advisable to consider other publications that consider the factors for identifying infective endocarditis with Staphylococcus aureus bacteraemia (for example, ref. 1-2, see below)

We have expanded upon our discussion and included the mentioned references as well (Page 8)

  1. The study noted no differences in mortality in groups with and without infective endocarditis. This is an interesting observation, but it has not been sufficiently studied by the authors. It is advisable to evaluate factors associated with mortality in patients with Staphylococcus aureus bacteraemia using univariate logistic regression.

We thank the Reviewer for this comment. We did not examine factors associated with mortality in our manuscript as it had not been the focus of our study. We had aimed to evaluate risk scores predicting IE in patients with SAB in a relatively ‘low-risk’ cohort off patients. Mortality was very low in our cohort overall, which is probably the reason why we could not establish a significant difference in outcomes between patients with IE and those who did.  Furthermore, we did not control or adjust for the treatment duration for these patients, as well as those who subsequently underwent surgery for the IE. When IE was appropriately identified and treated, it was possible that the mortality in this group may therefore be lower, and not too dissimilar from the group without IE.

We had described risk factors for mortality in in a previous abstract (https://doi.org/10.1093/ofid/ofac492.1485). As mentioned, mortality was low overall, which limited the degree of analyses in our cohort. Perhaps with longer prospective follow-up in future study, this may be the important subject of future study. We have included this important point in our limitations (Page 9).

References:

  1. Heriot GS, Cronin K, Tong SYC, Cheng AC, Liew D. Criteria for Identifying Patients With Staphylococcus aureus Bacteremia Who Are at Low Risk of Endocarditis: A Systematic Review. Open Forum Infect Dis. 2017 Nov 24;4(4):ofx261. doi: 10.1093/ofid/ofx261.

2.Khan UA, Zaidi SH, Majeed H, Lopez E, Tofighi D, Andre P, Schevchuck A, Garcia ME, Sheikh AB, Raizada V, Sheikhar R, Sagheer S. Clinical Risk Factors for Infectious Endocarditis Patients With Staphylococcus Aureus Bacteremia and the Diagnostic Utility of Transesophageal Echocardiogram. Curr Probl Cardiol. 2022 Nov;47(11):101331. doi: 10.1016/j.cpcardiol.2022.101331.

We thank the Editor and Reviewers for the kind and helpful comments. We hope the paper is now suitable for publication in the Journal.

Reviewer 2 Report

Comments and Suggestions for Authors

This is a single center retrospective study of the incidence and predictive factors for infective endocarditis amongst hospitalised patients with positive blood cultures for S.Aureus.

Whilst the study is interesting in principle (as many patients with S.Aureus are almost assumed to have IE until proven otherwise without this being appropriate for all), there are some major limitations in the interpretation of the results. Firstly, the follow up period of this cohort is very vague and the result seem to only pertain to one index hospitalisation without it being clear how long this was for, whether the bacteraemia was a recurrent issue causing hospitalisation, whether the patient had already had antibiotics or whether IE was considered equivocal on TEE and needed further clarification with serial TEE or follow up. 
All of these issues would need to be clarified for the article to be of scientific value to readers. 

Author Response

Ref.:  Ms. No. jcm-2920608

Title: Performance of risk scores in predicting infective endocarditis in patients with Staphylococcus aureus bacteraemia in a prospective Asian cohort

Journal of Clinical Medicine

We thank the Editor for allowing us the opportunity to revise our manuscript and the Reviewers for the important and constructive comments.  We have amended our paper in order to address the points raised by the Reviewer.

In the sections below, each of the points raised is identified and addressed with changes in the revised manuscript.

REVIEWER 2:

This is a single center retrospective study of the incidence and predictive factors for infective endocarditis amongst hospitalised patients with positive blood cultures for S.Aureus.

Whilst the study is interesting in principle (as many patients with S.Aureus are almost assumed to have IE until proven otherwise without this being appropriate for all), there are some major limitations in the interpretation of the results. Firstly, the follow up period of this cohort is very vague and the result seem to only pertain to one index hospitalisation without it being clear how long this was for, whether the bacteraemia was a recurrent issue causing hospitalisation, whether the patient had already had antibiotics or whether IE was considered equivocal on TEE and needed further clarification with serial TEE or follow up.

All of these issues would need to be clarified for the article to be of scientific value to readers.

We thank the Reviewer for these insightful comments. Indeed, our findings are exploratory. We examined the development of IE during the index admission for SAB, and have included the length of stay (on average ~23 days). We did not capture the development of IE beyond the index admission, but have included data on recurrence of bacteraemia and readmission within a year, which was relatively low, and comparable across the groups (Table 1).

We had not been able to capture the use of prior antibiotics or pre-treatment with antibiotics prior to admission, and agree that this may be an important consideration that may ‘mask’ the diagnosis of IE, by resulting in negative blood cultures. However, it is important to note that all our patients had positive blood cultures for S. aureus initially. Nevertheless, we also agree that it is possible that a proportion of patients may have had an initial echocardiographic imaging that may have been equivocal for IE as a result. If clinically indicated, these patients would have proceeded with serial imaging for the diagnosis of IE. As this was not a randomized controlled trial but a prospective cohort of patients with SAB, we had left the decision for serial imaging to the discretion of the managing physicians. If serial imaging resulted in the confirmed diagnosis of IE within the index hospitalisation, they were classified under the IE arm of the cohort. These are important considerations that we have since included this in our limitations (Page 9)

We thank the Editor and Reviewers for the kind and helpful comments. We hope the paper is now suitable for publication in the Journal.